# Drought adaptation in *Arabidopsis thaliana* by extensive genetic loss-of-function

**J Grey Monroe**[1,2]*, **Tyler Powell**[1,3], **Nicholas Price**[1], **Jack L Mullen**[1], **Anne Howard**[1], **Kyle Evans**[1], **John T Lovell**[4], **John K McKay**[1,2]

[1]Department of Bioagricultural Sciences and Pest Management, Colorado State University, Fort Collins, United States; [2]Graduate Degree Program in Ecology, Colorado State University, Fort Collins, United States; [3]Department of Biology, Colorado State University, Fort Collins, United States; [4]HudsonAlpha Institute for Biotechnology, Huntsville, United States

**Abstract** Interdisciplinary syntheses are needed to scale up discovery of the environmental drivers and molecular basis of adaptation in nature. Here we integrated novel approaches using whole genome sequences, satellite remote sensing, and transgenic experiments to study natural loss-of-function alleles associated with drought histories in wild *Arabidopsis thaliana*. The genes we identified exhibit population genetic signatures of parallel molecular evolution, selection for loss-of-function, and shared associations with flowering time phenotypes in directions consistent with longstanding adaptive hypotheses seven times more often than expected by chance. We then confirmed predicted phenotypes experimentally in transgenic knockout lines. These findings reveal the importance of drought timing to explain the evolution of alternative drought tolerance strategies and further challenge popular assumptions about the adaptive value of genetic loss-of-function in nature. These results also motivate improved species-wide sequencing efforts to better identify loss-of-function variants and inspire new opportunities for engineering climate resilience in crops.

DOI: https://doi.org/10.7554/eLife.41038.001

*For correspondence:
greymonroe@gmail.com

**Competing interests:** The authors declare that no competing interests exist.

## Introduction

Discovering the environmental drivers and functional genetics of adaptation in nature is a key goal of evolutionary biology and valuable to advance applied genetics in agriculture. Understanding the genetics of drought adaptation in plants is particularly important as crop losses resulting from droughts affect billions of people each year, posing the greatest threat to global food stability. Because droughts also impose strong selection on natural plant populations, investigating drought adaptation in wild species is both useful for addressing fundamental questions of evolutionary biology, such as determining whether adaptation proceeds by few or many alleles, and informative for efforts to reverse engineer drought tolerance in crops (*Mickelbart et al., 2015*). Such an evolutionary research program is motivated by the need to understand adaptive drought tolerance strategies for different types of drought conditions, which can vary in severity and timing (*Tardieu, 2012*). Furthermore, previous limitations of single gene approaches have reinforced the necessity of developing methods to identify beneficial alleles at genomic scales and functional molecular resolutions (*Dean and Thornton, 2007*; *Passioura, 2010*).

Drought stress can occur throughout the year and drought timing is forecast to change over the next century (*Trenberth et al., 2014*). While dramatic evolutionary responses to drought events have been documented, (e.g. *Franks et al., 2007*), little is known about the relationship between

**eLife digest** Water shortages caused by droughts lead to crop losses that affect billions of people around the world each year. By discovering how wild plants adapt to drought, it may be possible to identify traits and genes that help to improve the growth of crop plants when water is scarce. It has been suggested that plants have adapted to droughts by flowering at times of the year when droughts are less likely to occur. For example, if droughts are more likely to happen in spring, the plants may delay flowering until the summer.

*Arabidopsis thaliana* is a small plant that is found across Eurasia, Africa and North America, including in areas that are prone to drought at different times of the year. Individual plants of the same species may carry different versions of the same gene (known as alleles). Some of these alleles may not work properly and are referred to as loss-of-function alleles. Monroe et al. investigated whether *A. thaliana* plants carry any loss-of-function alleles that are associated with droughts happening in the spring or summer, and whether they are linked to when those plants will flower.

Monroe et al. analyzed satellite images collected over the last 30 years to measure when droughts have occurred. Next, they searched genome sequences of *Arabidopsis thaliana* for alleles that might help the plants to adapt to droughts in the spring or summer. Combining the two approaches revealed that loss-of-function alleles associated with spring droughts were strongly predicted to be associated with the plants flowering later in the year. Similarly, loss-of-function alleles associated with summer droughts were predicted to be associated with the plants flowering earlier in the year.

These findings support the idea that plants can adapt to drought by changing when they produce flowers, and suggest that loss-of-function alleles play a major role in this process. New techniques for editing genes mean it is easier than ever to generate new loss-of-function alleles in specific genes. Therefore, the results presented by Monroe et al. may help researchers to develop new varieties of crop plants that are better adapted to droughts.
DOI: https://doi.org/10.7554/eLife.41038.002

drought timing and adaptation. However, the observation both in nature and agriculture that plants are particularly susceptible to drought while flowering (*Nam et al., 2001*; *Dietrich and Smith, 2016*) has contributed to the longstanding hypothesis that adaptive flowering time should reflect patterns in the seasonal timing of drought events (*Passioura, 1996*). Detailed studies of life history also reveal that locally adapted *Arabidopsis thaliana* (*Arabidopsis* hereafter) populations begin flowering in their home environments just prior to and after periods of increased historical drought frequency (*Mojica et al., 2016*).

Flowering time in *Arabidopsis* is correlated with other drought tolerance traits such as water use efficiency and can serve as a proxy for alternative drought tolerance strategies, with early flowering genotypes being associated with low water use efficiency (drought escape strategy) and late flowering genotypes with high water use efficiency (dehydration avoidance strategy) (*McKay et al., 2003*; *Lovell et al., 2013*; *Kenney et al., 2014*). Thus, the historical timing of drought experienced by locally adapted populations may explain the evolution of these strategies and the distribution of alleles responsible for natural flowering time variation. This hypothesis motivated our investigation to identify alleles associated with drought timing and test the prediction that they contribute to adaptive flowering time evolution.

Identifying functionally relevant genetic variation contributing to adaptation is needed to understand fundamental evolutionary processes. In contrast to early theoretical predictions and popular assumptions, loss-of-function (LoF) alleles, those that eliminate or 'knockout' a gene's molecular function, are overrepresented among alleles reported as responsible for crop improvement and often produce adaptive phenotypes in wild species (*Hoekstra et al., 2006*; *Rausher, 2008*; *Olsen and Wendel, 2013*; *Alonso-Blanco and Méndez-Vigo, 2014*; *Weigel and Nordborg, 2015b*; *Torkamaneh et al., 2018*). Indeed, a number of individual genes exhibiting evidence of locally adaptive loss-of-function have been documented in *Arabidopsis* (*Grant et al., 1998*; *Johanson et al., 2000*; *Kliebenstein, 2001*; *Kroymann et al., 2003*; *Mouchel et al., 2004*; *Aukerman, 1997*;

*Hauser et al., 2001*; *Mauricio et al., 2003*; *Alonso-Blanco et al., 2005*; *Werner et al., 2005*; *Barboza et al., 2013*; *Xiang et al., 2014*).

Discovering adaptive LoF alleles is particularly valuable for inspiring targeted molecular breeding because functionally similar mutations can be mined from the breeding pool or generated directly by non-transgenic native gene editing. Unfortunately, traditional genome-wide association scans based on the one-locus two-allele model perform poorly at detecting adaptive LoF alleles, which because of the large number of mutations that can create them, are likely to arise through parallel molecular evolution (*Pennings and Hermisson, 2006*; *Barboza et al., 2013*; *Kerdaffrec et al., 2016*). Species-wide whole genome sequences however, present the opportunity to advance beyond previous mapping and scanning methods that relied on linked polymorphisms by instead characterizing and contrasting functionally defined alleles.

Here, we combined long-term satellite-detected drought histories, whole genome sequence scans based on allele function, and transgenic knockout experiments in *Arabidopsis* to test historical predictions about how drought timing shapes the evolution of flowering time and outline a broadly scalable approach for discovering loss-of-function gene variants contributing to plant climate adaptation.

## Results and discussion

To study global seasonal drought timing, satellite-detected measurements offer a valuable historical record. One such measurement, the Vegetative Health Index (VHI) has been used for decades to monitor drought, including in many places across the natural range of *Arabidopsis* (*Kogan, 1997*). Though primarily used as a tool to predict crop productivity, by quantifying drought induced vegetative stress this index also provides a resource for evolutionary ecologists to study seasonal patterns in drought-related episodes of natural selection. We analyzed 34 years of VHI data to characterize drought regimens at the home environments of *Arabidopsis* ecotypes (*Figure 1*, *Supplementary file 1*). We found that drought frequency during the spring (ß = 50.016, $p < 2 \times 10^{-16}$) and summer (ß = $-28.035$, p = $4.4 \times 10^{-7}$) significantly predict flowering time among *Arabidopsis* ecotypes (*Supplementary file 2A*). We then generated a drought-timing index that quantifies the relative frequency of drought between spring and summer over the typical reproductive growing season and observed substantial differences in drought timing experienced by ecotypes (*Figure 1—figure supplement 1*). This environmental variation presented a useful cline to address classical hypotheses about the evolution of flowering time in relation to drought timing and identify LoF alleles potentially contributing to this evolution.

To identify candidate LoF alleles underlying drought adaptation and flowering time evolution, we analyzed whole genome sequences in *Arabidopsis*. We first surveyed the genomes of 1135 ecotypes (*1001 Genomes Consortium, 2016*) for LoF alleles in protein coding genes predicted to encode truncated amino acid sequences (*Supplementary file 3A*). To overcome the likely parallel evolutionary origins of LoF alleles that would have challenged previous methods, we classified alleles based functional allele state rather than individual polymorphisms for association testing. After filtering to reduce the likelihood of false positives (see materials and methods), we thus tested 2088 genes for LoF allele associations with drought timing (*Figure 2A*) and flowering time (*Figure 2B*). These analyses identified 247 genes in which LoF alleles are significantly associated with drought timing and/or flowering time after accounting for population structure and multiple testing (*Supplementary file 3B*). In contrast, when we performed these analyses on a permuted LoF genotype matrix, we found no genes that were significantly associated with drought timing or flowering time (*Figure 1—figure supplement 1*).

It should be noted that the 2088 genes tested for associations to flowering time and drought timing are not a complete representation of LoF alleles in *Arabidopsis*. In some cases, previously studied LoF alleles did not pass filtering steps (*Supplementary file 3D,E*). This was primarily because the frequency or quality of LoF allele calls in these genes fell below our filtering requirements (see materials and methods). In other cases, the Col-0 reference genome already has a documented LOF allele. Finally, we expect LoF alleles to be undetectable if they are the product of large insertions or deletions which cannot be properly identified with currently available resequencing data. Thus, while the methods used here are designed to minimize false positives (alleles classified as LoF, but which are actually functional), the likely occurrence of false negatives (undetected LoF alleles) in available

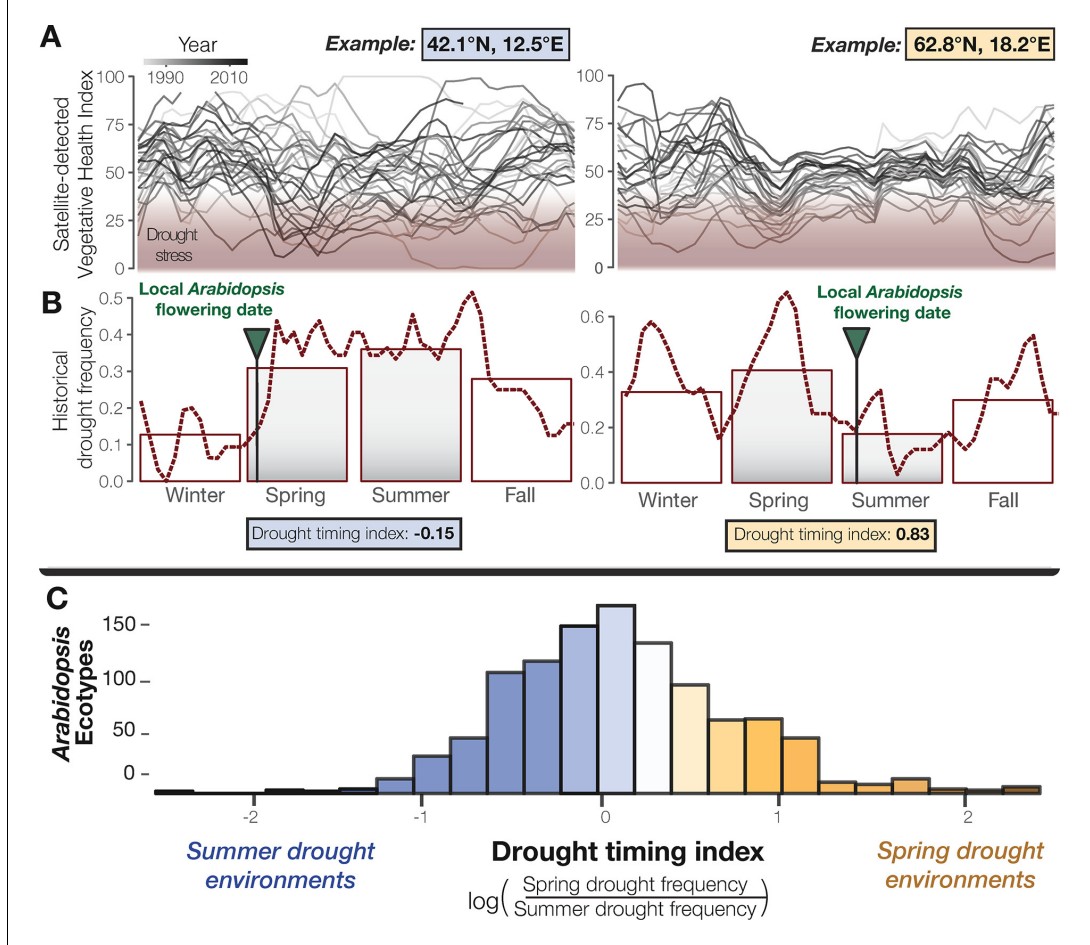

**Figure 1.** Seasonal drought timing varies across the *Arabidopsis* species range. (**A**) Examples of home environments for two well-studied *Arabidopsis* ecotypes (*Mojica et al., 2016*) from Italy and Sweden, left and right plots respectively, showing historical drought conditions detected using the VHI and (**B**) drought frequency (VHI <40, NOAA drought classification) by week (line) and season (bars). Arrows mark locally observed flowering dates (*Mojica et al., 2016*) and gray bars highlight the typical reproductive growing season used to quantify a drought-timing index. (**C**) Variation in historical drought timing experienced at the home environments of *Arabidopsis* ecotypes across the species range (figure supplement). Large values indicate environments where spring droughts occur more frequently than summer drought (i.e. where the frequency of drought decreases over the course of the typical reproductive growing season) and vice versa.

DOI: https://doi.org/10.7554/eLife.41038.003

The following figure supplement is available for figure 1:

**Figure supplement 1.** *Arabidopsis* ecotypes are distributed across satellite-detected drought timing gradients.

DOI: https://doi.org/10.7554/eLife.41038.004

data motivates the need for more sophisticated species wide genome sequencing efforts including a greater diversity of de-novo quality genomes for comprehensive detection of functionally relevant genetic variation across the species.

Associations to drought timing predicted associations of LoF alleles to flowering time directly. Together, summer drought and earlier flowering associated genes (*Figure 2C*), and spring drought and later flowering associated genes (*Figure 2D*) overlapped seven times more often than expected by chance ($\chi^2$=492, p < 2 × $10^{-16}$) and no shared associations were observed in the opposite direction. The strengths of the associations between LoF alleles and drought timing (P values) was also strongly correlated with the strengths of the associations to flowering time ($r^2$ = 0.48. *Figure 2—figure supplement 1E*, *Figure 2C,D*). This result is comparable to overlapping peaks in a 'Manhattan plot' generated from a traditional genome wide association scan (e.g. *Bosse et al., 2017*). In contrast, these associations were weakly correlated when genotypes were permuted ($r^2$ = 0.01

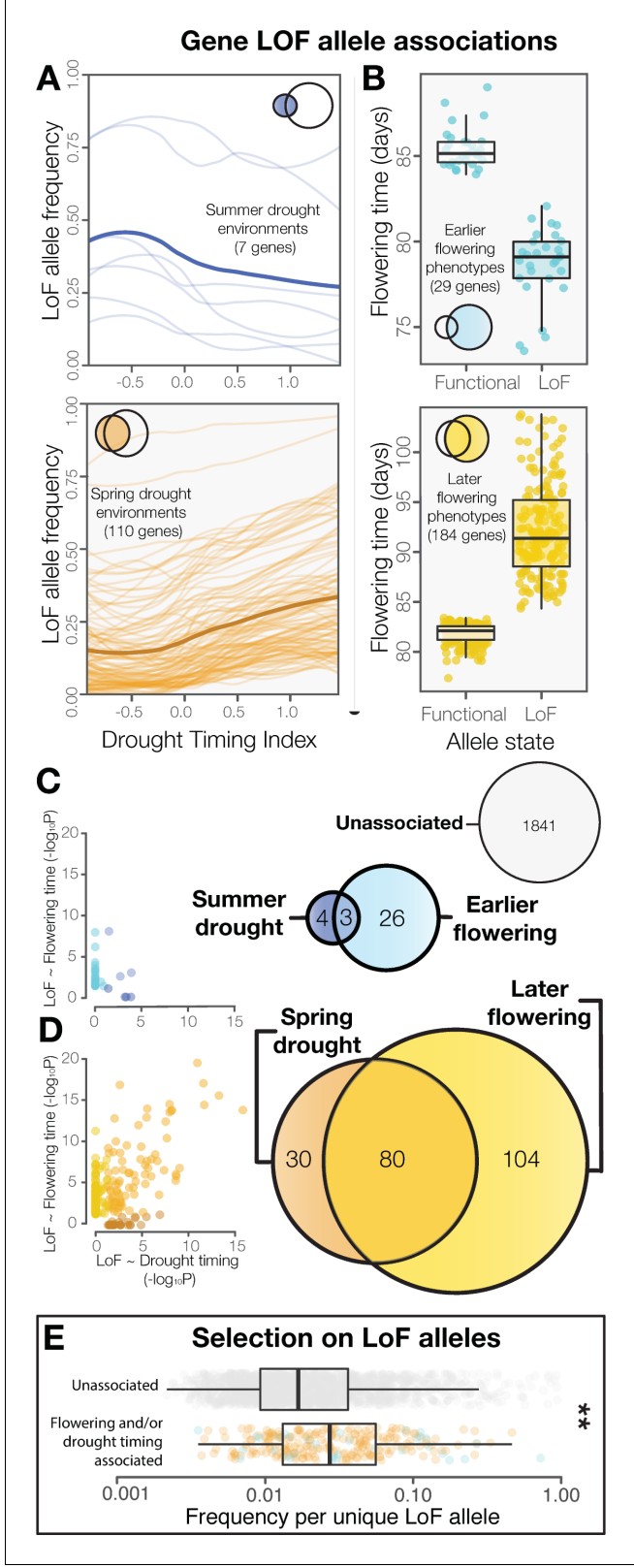

**Figure 2.** LoF alleles share associations between drought timing and flowering time, exhibit evidence of positive selection. (**A**) Visualization of the frequency of LoF alleles across environments in genes associated to summer (upper) or spring drought environments (lower). Darker lines indicate the mean across genes. (**B**) Contrasting flowering times between ecotypes with functional versus LoF alleles in genes associated with earlier (upper) or

*Figure 2 continued on next page*

*Figure 2 continued*

later (lower) flowering time phenotypes. (C) Overlap and relationships between the strength of LoF allele associations in genes associated with summer drought and earlier flowering, and (D) spring drought and later flowering. (E) Increased frequencies of independent LoF alleles in genes associated with drought timing and/or flowering time compared to genes without detected associations (t-test, p = 3.4 × 10$^{-7}$), a signature of recurrent mutation accompanied by positive selection (*Pennings and Hermisson, 2006*).

DOI: https://doi.org/10.7554/eLife.41038.005

The following figure supplements are available for figure 2:

**Figure supplement 1.** P values of LoF allele associations.

DOI: https://doi.org/10.7554/eLife.41038.006

**Figure supplement 2.** Signatures of selection on LoF genes identified differ from null expectations.

DOI: https://doi.org/10.7554/eLife.41038.007

**Figure supplement 3.** LoF alleles are not broadly overabundant in *Arabidopsis* ecotypes originating from spring drought environments or flowering later.

DOI: https://doi.org/10.7554/eLife.41038.008

---

*Figure 2—figure supplement 1F*), indicating that the result is not simply explained as an artifact of allele frequencies or by the relationship between drought timing and flowering time (i.e. *Supplementary file 1A*). Thus, satellite-detected drought histories and a functional genome-wide scanning approach prove useful for predicting the direction and molecular targets of phenotypic evolution. Similar investigations with ecologically meaningful environmental variation could be valuable for discovering candidates underlying other important traits that are especially difficult to measure.

These results further support the classical hypothesis that the relationship between phenology and drought timing is the most important feature of plant drought tolerance (*Passioura, 1996*), indicating the evolution of 'drought escape' through earlier flowering in summer drought environments, and 'dehydration avoidance' by later flowering genotypes in spring drought environments. Because most *Arabidopsis* populations appear to exhibit a winter annual life habit, germinating in the fall and overwintering as a rosette (*Ratcliffe, 1961*; *Thompson, 1994*; *Burghardt et al., 2015*), late flowering genotypes in spring drought environments are expected to still encounter drought conditions. However, delayed flowering may ensure that droughts co-occur with vegetative growth rather than during the drought sensitive reproductive phase. This pattern is also consistent with hypotheses explaining the more water conservative water use and stomatal traits observed in late flowering genotypes (*McKay et al., 2003*; *Lovell et al., 2013*; *Kenney et al., 2014*; *Kooyers, 2015*) and those from spring drought environments (*Dittberner et al., 2018*). Future experimental work will be valuable to identify other plant physiological traits affected by the LoF alleles associated with drought timing.

These results provide new insight into the ecology and genetics of *Arabidopsis* life history evolution, but the complex ecological reality of these processes is undoubtedly beyond the scope of this study. We found that drought timing remains a significant predictor of allele associations to flowering time when controlling for allele associations with latitude and minimum temperature (slope estimate in multiple linear regression, p < 2×10$^{-16}$, *Supplementary file 2B*). However, other unknown climatic variables or environmental interactions and non-linearities likely contribute to the flowering time adaptation as well. Flowering time is only one component of phenology and other adaptive life history transitions such a germination timing (*Donohue, 2002*) may also be influenced by drought timing and could change how drought timing affects the evolution of flowering time, a hypothesis that warrants further investigation. Furthermore, measuring flowering time in other environments, such alternate light regimes, may yield a different set of candidate genes using similar approaches.

Signatures of selection in the genes identified differ from the genome average and neutral expectations. As expected for genes harboring LoF alleles, these show parallel evolution of LoF and accelerated amino acid sequence evolution among *Arabidopsis* ecotypes (*Figure 2—figure supplement 2A,B*, *Supplementary file 2C*). We also found evidence of positive selection for LoF alleles in genes associated with drought timing and/or flowering time. While these genes have similar global frequencies of LoF alleles compared to genes not showing associations with drought timing and/or flowering time (*Figure 2—figure supplement 2C*), they tend to have significantly fewer unique LoF

alleles (*Figure 2—figure supplement 2D*) and greater frequencies of each independent LoF allele (*Figure 2E*). This pattern is consistent with theoretical predictions and results from simulations of adaptation by parallel molecular evolution involving recurrent mutation combined with more rapid local fixation of alleles experiencing positive selection (*Pennings and Hermisson, 2006*). In cases where adaptation proceeds through the fixation of a single adaptive allele, traditional genome scanning approaches may be sufficient to detect causal loci. However, when genetic variation consists of multiple independent alleles, as is often the case for the genes examined here (*Figure 2—figure supplement 2D*), classifying alleles functionally before testing for associations is likely necessary.

The extent of LoF responsible for adaptive phenotypic evolution is much greater than once assumed (*Smith, 1970*; *Albalat and Cañestro, 2016*). LoF alleles identified were overwhelmingly associated with spring drought or later flowering rather than summer drought or earlier flowering ($\chi^2$ = 132, p < 2 × 10$^{-16}$, *Figure 2*). Because the reference genome and gene models are from an early flowering *Arabidopsis* line, Col-0, this is consistent with the hypothesis that LoF alleles are particularly important in the evolution of phenotypic divergence (*Rausher, 2008*). This result also highlights the need to develop functional genomics resources informed by multiple de-novo quality reference genomes. We found that flowering time is strongly predicted by the accumulation of LoF alleles across the 214 candidate genes associated to spring drought and/or later flowering time

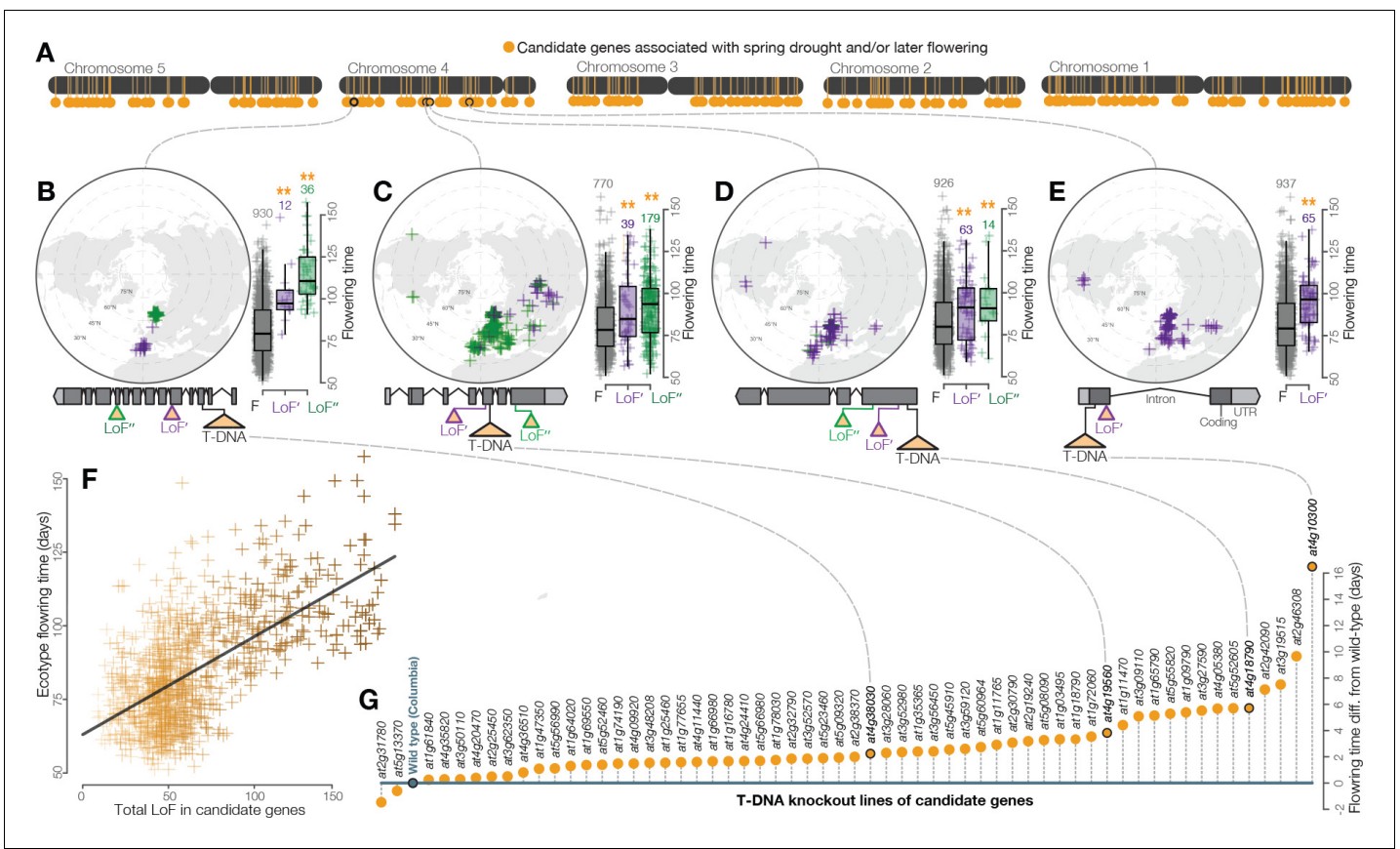

**Figure 3.** Widespread LoF contributing to later flowering time evolution. (A) Genomic map of 214 candidate genes with associations between LoF alleles and spring drought environments and/or later flowering time phenotypes. (B–E) Examples of the geography and flowering times among *Arabidopsis* ecotypes of LoF alleles in candidate genes including; (B) a previously unstudied rhamnogalacturonate lyase, (C) a cyclin linked to later flowering in prior knockout experiments (*Cui et al., 2007*), (D) members of the drought-responsive Nramp2 (*Qin et al., 2017*) (E) and RmlC-like cupin (*Aghdasi et al., 2012*) protein families. (F) Later flowering time in ecotypes predicted by the accumulation of LoF alleles across all candidate genes. The line shows the best fitting model. Color scale of points reflects proportion of total LoF in ecotypes that are candidate genes (darker points = greater proportion) (G) Experimental validation of hypothesized later flowering time in T-DNA knockout lines of candidate genes compared to the wild type genotype.

DOI: https://doi.org/10.7554/eLife.41038.009

(*Figure 3A–E*), estimating a 1 day increase for every three additional LoF alleles across these candidate genes (*Figure 3F*). This relationship is best represented as a simple linear regression; the addition of a non-linear quadratic predictor variable did not significantly improve the fit of the model (F = 0.7005, p = 0.4028). Importantly, we did not find a broader overabundance of LoF alleles in later flowering ecotypes or those from spring drought environments that would explain this relationship (e.g. *Figure 2—figure supplement 3*). Rather, these findings support a model of climate-associated evolution in complex traits that includes a substantial contribution from widespread genetic LoF and give promise to targeted LoF for directed phenotypic engineering.

Experimental knockout lines confirmed the later flowering times predicted from natural allele associations. To test phenotypic effects, we screened a panel of confirmed T-DNA insertion mutants representing a sample of candidate LoF alleles associated with spring drought and/or later flowering. As predicted by variation among *Arabidopsis* ecotypes (*Figure 2D*), the vast majority of knockout lines in these candidate genes (57 of 59, $\chi^2$ = 51, p = 8.045e-13) flowered later on average than the wild type genotype (*Figure 3G*, Supplementary file SF). LoF alleles identified through these analyses and experiments include those previously linked to flowering time (*Cui et al., 2007*) and drought responses (*Aghdasi et al., 2012*; *Qin et al., 2017*). Implementing a functional genome-wide association scan, we find that allele associations with ecologically meaningful environmental variation (drought timing) accurately predict associations with adaptive phenotypes directly (flowering time).

Together with validation in transgenic lines, these findings outline a scalable model for gaining deeper insights into the functional genomics of climate adaptation in nature. Combining large scale knockout experiments with functional genome wide association scans may be a valuable approach for future research to quantify the power to predict LoF allele effects. These results also further challenge historical assumptions about molecular adaptation that have implications for influencing evolutionary theory and public attitudes toward emerging molecular breeding approaches.

Groundbreaking yield increases during the green revolution of the 1960 s were largely attributable to semi-dwarf phenotypes caused by LoF alleles in both rice and barley (*Spielmeyer et al., 2002*; *Jia et al., 2009*). Later it was found that natural LoF alleles of the same gene in wild *Arabidopsis* produce similar phenotypes (*Barboza et al., 2013*), suggesting the potential to mine ecological species for information directly useful for crop improvement. Visions of a second green revolution powered and informed by such natural variation call for discoveries in evolutionary functional genomics at scales that have now become possible. The genes identified here could inspire future molecular breeding of climate resilient crops and this work more broadly highlights the value of integrating diverse disciplines to scale up the discovery of the climatic drivers of adaptation and functionally significant genetic variation at molecular resolutions.

## Materials and methods

### Satellite-Detected drought histories of *Arabidopsis*

To study patterns in historical drought, the remotely sensed Vegetative Health Index (VHI) was used, a satellite-detected drought measurement tool whose advantage is that it includes information about vegetative impacts of drought (*Passioura, 1996*; *AghaKouchak et al., 2015*). This index is based on multiple data sources from NOAA satellites, combining deviations from historic climatic (Temperature Condition Index derived from AVHRR-based observations in thermal bands) and vegetative conditions (Vegetative Condition Index derived from NDVI) to detect periods of ecological drought conditions and distinguish between other sources of vegetative stress such as cold (*Kogan, 1997*; *Kogan et al., 2005*; *Rojas et al., 2011*). VHI was collected weekly since 1981 at 16 $km^2$ resolution on a scale from 0 to 100, where values below 40 reflect drought conditions (*Kogan, 1997*) (*Figure 1A*). The frequencies of observing drought conditions during photoperiodic spring (quarter surrounding spring equinox), summer (quarter surrounding summer solstice), fall (quarter surrounding fall equinox), and winter (quarter surrounding winter solstice) were calculated globally from 1981 to 2015 (*Figure 1B*) in R (*R Core Development Team, 2017*) using the *raster* package (*Hijmans, 2016*).

After removing ecotypes with missing location data or locations falling within pixels classified as water, seasonal drought frequencies and drought timing were calculated at the location of origin for

1,097 *Arabidopsis* ecotypes that were included as part of the 1001 Genomes Project (*1001 Genomes Consortium, 2016*) (*Figure 1C*, *Supplementary file 1*). Up to date global map files of seasonal drought frequency and the drought-timing index used here are available on Dryad and greymonroe. github.io/data alongside a brief tutorial showing how to extract data for points of interest in R. We tested whether seasonal drought frequencies significantly predicted with flowering time (flowering time described in subsequent section regarding LoF associations) by multiple linear regression (*Supplementary file 2A*)

To characterize the seasonal timing of droughts during an important period of *Arabidopsis'* life history, a univariate drought-timing index was generated that quantifies whether the historical frequency of drought increases or decreases over the course of the typical *Arabidopsis* reproductive growing season (*Ratcliffe, 1961*; *Thompson, 1994*; *Burghardt et al., 2015*). Specifically, this index is equal to the natural log transformed ratio between spring and summer drought frequency. More negative values reflect environments where drought frequency increases from spring to summer and are referred to here as 'summer drought environments,' (e.g. *Figure 1B* left). Conversely, more positive values reflect environments where drought frequency decreases from spring to summer and are referred to here as 'spring drought environments,' (e.g. *Figure 1B* right).

## Loss-of-Function (LoF) Alleles in *Arabidopsis* genomes

To identify functionally definitive gene variants (*Hoekstra and Coyne, 2007*; *Weigel and Nordborg, 2015a*; *Byers et al., 2017*), LoF alleles (*Albalat and Cañestro, 2016*) were identified from whole genome sequence data of 1,135 *Arabidopsis* accessions (*Olson, 1999*; *Cutter and Jovelin, 2015*; *1001 Genomes Consortium, 2016*) using R scripts. First, genes were filtered to those containing at least 5% frequency of predicted frameshift or premature stop mutations and less than 5% missing allele calls from results generated by the 1,001 Genomes Consortium (*1001 Genomes Consortium, 2016*) using 'SnpEff' (*Cingolani et al., 2012*). To reduce instances where exon skipping might ameliorate LoF mutations (*Gan et al., 2011*), genes were filtered to those with a single predicted gene model (*Lamesch et al., 2012*). Additionally, to preclude false LoF calls for cases where compensatory mutations restore gene function or in which an insignificant portion of the final protein product is affected by putative LoF mutations (*MacArthur et al., 2012*), coding regions were translated into predicted amino acid sequences from which lengths from start to stop codon were calculated in R. LoF alleles were defined as those producing protein products with at least 10% lost because of late start codons and/or prematurely truncated translation. Allelic heterogeneity expected to mask these genes from traditional GWAS (*Remington, 2015*; *Monroe et al., 2016*; *Flood and Hancock, 2017*) was corrected for by classifying all alleles as either functional (0) or non-functional (1). A final frequency filter was re-applied (5% global LoF allele frequency), resulting in 2088 genes for downstream association analyses (*Supplementary file 3B*). Finally, to compare the results of this pipeline to genes known to harbor natural LoF alleles (*Mouchel et al., 2004*; *Shindo et al., 2008*; *Gujas et al., 2012*; *Kliebenstein, 2001*; *Kroymann et al., 2003*; *Grant et al., 1998*; *Tian et al., 2003*; *Mauricio et al., 2003*; *Werner et al., 2005*; *Aukerman, 1997*; *Flowers et al., 2009*; *Xiang et al., 2014*; *Xiang et al., 2016*; *Amiguet-Vercher et al., 2015*; *Johanson et al., 2000*; *Le Corre et al., 2002*; *McKay et al., 2003*; *Stinchcombe et al., 2004*; *Shindo et al., 2005*; *Flowers et al., 2009*; *Méndez-Vigo et al., 2011*; *Lovell et al., 2013*; *Hauser et al., 2001*; *Bloomer et al., 2012*; *Alonso-Blanco et al., 2005*; *Zhen and Ungerer, 2008*; *Kang et al., 2013*; *Monroe et al., 2016*; *Zhu et al., 2015*; *Barboza et al., 2013*), we manually performed this functional allele calling approach on a set of 16 genes (Supplementary file D,E)

## LoF associations to drought timing and flowering time

To identify candidate LoF alleles responsible for climate adaptation and phenotypic evolution, the relationships between functional allele state and drought timing and between functional allele state and flowering time were evaluated for each of the 2088 genes that passed preceding filtering steps. Specifically, the association between functional allele state among *Arabidopsis* ecotypes and historical drought timing at their locations of origin was tested by logistic regression in a generalized linear model in R (*R Core Development Team, 2017*). This association study differs from traditional GWAS in several respects. First, because the alleles studied here are functionally defined, they are expected to be more likely to have a phenotypic impact than random SNPs. Second, the scope of

our analyses were restricted to a subset of the genome - 2088 genes with high confidence LoF allele calls that passed previous filtering steps, rather than tens of thousands to millions of SNPs. Finally, in contrast to traditional GWAS, which is designed to identify associated chromosomal regions rather than functionally definitive genetic variations, our approach is motivated by the ability to identify alleles at molecular resolutions whose functional relevance can be tested empirically. Thus, the balance of opportunity costs related to trade-offs between false positive and false negative associations that generally challenge GWAS are shifted to reduce false negatives rather than minimizing false positives. For these reasons, we implemented analyses based on (*Price et al., 2006*) to balance false positives and false negatives. Population structure was accounted for by performing a principal component analysis on the kinship matrix among all ecotypes and including in each model the first three resulting principal components, which explain >75% of variance in relatedness between ecotypes (*Price et al., 2006*). The P-values ($P_{drought\ timing}$) of the slope estimates ($\beta_{drought\ timing}$) for drought timing in these models were adjusted to account for multiple tests by a Bonferroni correction to identify those significantly associated (*Supplementary file 3C*).

Summer drought genes were identified as those in which LoF alleles are found in ecotypes that experience a significantly ($\beta_{drought\ timing} < 0$ and $P_{drought\ timing} < 0.05$) more negative drought-timing index (summer drought environments where drought frequency increases over the course of the reproductive growing season, *Figure 1B* left and *Figure 2A* top). Conversely, spring drought genes were identified as those in which LoF alleles are found in ecotypes that experience a significantly ($\beta_{drought\ timing} > 0$ and $P_{drought\ timing} < 0.05$) more positive drought-timing index (spring drought environments where drought frequency decreases over the course of the reproductive growing season, *Figure 1B* right and *Figure 2A* bottom).

The above analytical approach was repeated to test whether functional allele state is associated with the reported common garden flowering times of *Arabidopsis* ecotypes (*Alonso-Blanco and Méndez-Vigo, 2014*) (*Supplementary file 1*). See Alonso-Blanco *et al.* (*Alonso-Blanco and Méndez-Vigo, 2014*) for details, but in brief, flowering time was measured in growth chambers at 10°C (considerably less missing data than experiment at 16°C) under 16 hour days. Earlier flowering genes were identified as those in which LoF alleles are found in ecotypes that flower significantly ($\beta_{flowering\ time} < 0$ and $P_{flowering\ time} < 0.05$) earlier than ecotypes with a functional allele (*Figure 2B* top). Later flowering genes were identified as those in which LoF alleles are found in ecotypes that flower significantly ($\beta_{flowering\ time} > 0$ and $P_{flowering\ time} < 0.05$) later than ecotypes with a functional allele (*Figure 2B* bottom). The preceding analyses revealed considerable overlap between genes associated with both drought timing and flowering time. To assess whether this result was an artifact of the binary LoF allele calls, we randomly permuted the genotype matrix and repeated the analyses described above, testing for significant associations between allele states and drought timing and/or flowering time. Quantile-quantile plots of P values were visualized using qqPlot in the GWASTools package in R (*Gogarten et al., 2012*) (*Figure 2—figure supplement 1A–D*)

## Overlap between drought timing and flowering time associated genes

To address the longstanding hypothesis that flowering time reflects adaptation to drought timing (*Fox, 1990*; *Passioura, 1996*; *Kooyers, 2015*), and to test the corresponding prediction that alleles associated with drought timing are also associated with flowering time, the groups of genes identified with significant associations to drought timing or flowering time were compared (*Figure 2C and D*). Deviation from the null hypothesis of independent associations to drought timing and flowering time was evaluated by a chi-squared test (Expected number of co-associated genes = 12, Observed = 83, $\chi^2 = 492$, p = $2 \times 10^{-16}$).

The magnitude of P-values have historically served as the basis of selecting candidate loci for further examination toward their contribution to environmental adaptation or phenotypic evolution in quantitative trait locus mapping and genome wide association scans [e.g. (*Bosse et al., 2017*). To test whether associations to environment (drought timing) can be used to identify loci associated with phenotypes (flowering time) directly, the correlation between log transformed P-values describing allele associations with drought timing ($P_{drought\ timing}$) and with flowering time ($P_{flowering\ time}$) was calculated (*Figure 2—figure supplement 1E*, $r^2 = 0.48$,) and visualized separately for genes associated to summer drought/earlier flowering (*Figure 2C*) and to spring drought/later flowering (*Figure 2D*). To control for the possibility that allele frequencies or the relationship between drought timing and flowering time explained these observations, we also tested whether allele associations

were correlated when generated from association analyses using a matrix of randomly permuted genotypes with the same allele frequencies (*Figure 2—figure supplement 1F*, $r^2$ = 0.01).

Finally, to control for the possibility that correlated LoF allele associations were explained by confounding environmental variables we tested whether the LoF allele associations to drought timing remained predictive while accounting for LoF allele associations with latitude and minimum temperature of the coldest month (*Hijmans et al., 2005*) using a multiple linear regression in R (*Supplementary file 3B*). To do so, we repeated the association analyses described in the previous section but instead tested for LoF allele associations with latitude and minimum temperatures. We then included these P values (*Supplementary file 2B*) in a multiple linear regression where the strength of the association to flowering time was predicted by the associations to drought timing, latitude, and minimum temperature simultaneously.

## Signatures of selection

To assess whether histories of selection for genes identified differ from the genome wide expectation, measures of amino acid sequence evolution were evaluated for 122 genes in which loss-of-function is associated with drought timing or flowering time and for which there are orthologs identified between *A. lyrata* and *A. thaliana* (*Goodstein et al., 2012*). For each gene, sequences were aligned using MAFFT (*Katoh and Standley, 2013*), codons with gaps removed, and the number of non-synonymous and synonymous polymorphisms among *A. thaliana* accessions ($P_N$ and $P_S$) as well as synonymous and non-synonymous divergence ($D_N$ and $D_S$) from *A. lyrata* were measured using mkTest.rb (https://github.com/kern-lab/). The ratios $P_N/P_S$ and $D_N/D_S$ were then calculated to measure the proportion of variants predicted to affect amino acid sequences that are segregating among ecotypes and diverged from *A. lyrata*, respectively. These calculations were also performed for genes not associated to drought timing or flowering time (n = 912) and the remaining genes across the *A. thaliana* genome (n = 20373) with orthologs between *A. lyrata* and *A. thaliana*. To test whether genes identified show evidence of accelerated protein sequence evolution, comparisons were made to genes associated with drought timing or flowering time for both $P_N/P_S$ (*Figure 2—figure supplement 2A*) and $D_N/D_S$ ((*Figure 2—figure supplement 2A,B*) by two-sided students t-tests (α = 0.05) in R (*R Core Development Team, 2017*).

Because theory predicts adaptation by loss-of-function to proceed through multiple independent alleles, but to exhibit a fewer number of different alleles than in neutral loci at similar LoF allele frequencies (*Pennings and Hermisson, 2006*; *Ralph and Coop, 2010*; *Ralph and Coop, 2015*), the number of unique LoF alleles was estimated by protein length in the genes that passed preceding filtering steps. To address the hypothesis that genes in which LoF alleles are associated to drought history or flowering time are likely to reflect positive selection compared to genes in which LoF are random with respect to drought history or flowering time, the total number of unique LoF alleles between these groups was compared using a two-sided students t-test ($log_{10}$ transformed, p = 5.8×10$^{-7}$, (*Figure 2—figure supplement 2D*). To control for the possibility that this result in an artifact of reduced frequency of LoF alleles in genes identified, the global frequency of LoF was also compared between these groups ($log_{10}$ transformed, two-sided students t-test, p = 0.11, (*Figure 2—figure supplement 2C*). Finally, to further test the prediction that LoF alleles in genes identified have increased in frequency because of more positive selection, the frequency per specific LoF allele was compared between groups ($log_{10}$ transformed, two-sided students t-test, p = 3.4×10$^{-7}$, *Figure 2E*).

## Candidate genes contributing to later flowering time by widespread LoF

The significance of the tendency for LoF associations to spring drought/later flowering time (*Figure 2D*) was tested by chi-squared tests (spring drought vs. summer drought, p < 2×10$^{-16}$; later vs. earlier flowering, p < 2×10$^{-16}$, spring drought/later flowering vs. summer drought/earlier flowering, p < 2×10$^{-16}$). The chromosomal locations of candidate genes (those associated to spring drought/later flowering time) were mapped onto the *Arabidopsis* genome (*Lamesch et al., 2012*) (*Figure 3A*). To address the hypothesis that widespread LoF contributes to later flowering time phenotypes, the total number of LoF in candidate genes for each ecotype was calculated and the correlation between this value and flowering time evaluated (*Figure 3F*, $r^2$ = 0.39, p < 2×10$^{-16}$). We also

tested whether a model which included a non-linear predictor (squared value of the total number of LoF in candidate genes) was a better fit than the simple linear model by an analysis of variance (F = 0.7005, p = 0.4028).

## Experimental testing of predicted phenotypes in gene knockout lines

The preceding analyses provided compelling evidence of LoF in candidate genes as important in the evolution of later flowering time phenotypes. To test the prediction that non-functionalization of these genes causes increased flowering time, phenotypes were measured in transgenic lines in a subsample of candidate genes showing a significant association between loss-of-function and spring drought environments and/or later flowering time. Motivated by the general need to develop a high throughput approach of studying naturally adaptive LoF, knockout lines from the Arabidopsis Biological Resource Center were chosen from a collection created by the SALK Institute in which a T-DNA insertion in an exon of candidate genes has already been identified and confirmed to be homozygous (*O'Malley and Ecker, 2010*; *Rutter et al., 2017*). These T-DNA knockout lines were generated by the SALK institute (*Supplementary file 3F*) and exist in a common genetic background (Columbia) (*Alonso et al., 2003*). Seeds were planted in 2' pots containing wet potting soil and stratified for 5 days at 4°C. Seedlings were thinned to a single plant per pot one week after stratification. Plants were grown (59 T-DNA knockout lines, 10 reps of each line and 30 reps Columbia) in a stratified (by shelf), randomized design in growth chambers (Conviron ATC60, Controlled Environments, Winnipeg, MB) under 16 hr of light at 20°C. Flowering time was measured as days after planting to the emergence of the first open flower, based on the definition of flowering time used by the 1,001 Genomes Consortium (*1001 Genomes Consortium, 2016*). We calculated the least squares mean (lsmean from 'lsmeans' package in R) flowering time for each line from a mixed model where shelf and tray were included as random effects (*Supplementary file 3F*). We tested the prediction that knockout lines would flower later (have higher lsmean flowering time estimates) than the wild type Columbia genotype by a chi-squared test (p = $8.1 \times 10^{-13}$).

## Acknowledgements

E Buckler, D Des Marais, A Henry, J Lasky, T Mitchell-Olds, J Ross-Ibarra, and D Sloan provided valuable feedback and insightful discussion that improved this work. This study was financially supported by NSF Awards DEB 1022196 and 1556262 to JKM, NSF Award 1701918 and USDA-NIFA Award 2014-38420-21801 to JGM, as well as generous funding from Cargill, Inc. The work conducted by the US Department of Energy Joint Genome Institute, a DOE Office of Science User Facility, is supported by the Office of Science of the U.S. Department of Energy under Contract No. DE-AC02-05CH11231. Data used are included in the main text, supplementary materials, and public repositories.

## Additional information

### Funding

| Funder | Grant reference number | Author |
| --- | --- | --- |
| National Science Foundation | 1701918 | John Grey Monroe |
| U.S. Department of Agriculture | 2014- 38420-21801 | John Grey Monroe |
| National Science Foundation | IOS-1402393 | John T Lovell |
| National Science Foundation | 1022196 | John K McKay |
| National Science Foundation | 1556262 | John K McKay |
| Cargill | Research support | John K McKay |

The funders had no role in study design, data collection and interpretation, or the decision to submit the work for publication.

## Author contributions

J Grey Monroe, Conceptualization, Data curation, Formal analysis, Funding acquisition, Validation, Investigation, Visualization, Methodology, Writing—original draft, Project administration, Writing—review and editing; Tyler Powell, Validation, Investigation, Methodology, Writing—review and editing; Nicholas Price, Data curation, Formal analysis, Methodology, Writing—review and editing; Jack L Mullen, Anne Howard, Supervision, Investigation, Methodology, Project administration, Writing—review and editing; Kyle Evans, Investigation, Methodology, Writing—review and editing; John T Lovell, Data curation, Formal analysis, Investigation, Methodology, Writing—review and editing; John K McKay, Conceptualization, Resources, Supervision, Funding acquisition, Methodology, Project administration, Writing—review and editing

## Author ORCIDs

J Grey Monroe (iD) https://orcid.org/0000-0002-4025-5572
Nicholas Price (iD) http://orcid.org/0000-0001-6672-2952
Kyle Evans (iD) http://orcid.org/0000-0002-9895-5010
John T Lovell (iD) https://orcid.org/0000-0002-8938-1166

## Decision letter and Author response

Decision letter https://doi.org/10.7554/eLife.41038.020
Author response https://doi.org/10.7554/eLife.41038.021

# Additional files

## Supplementary files

• Source data 1. Raw flowering time measurements for of wild-type genomic background and T-DNA knockout lines.
DOI: https://doi.org/10.7554/eLife.41038.010

• Supplementary file 1. *Arabidopsis* ecotypes examined. Includes ecotype identifiers as well as latitude and longitude of origin, seasonal drought frequencies (winter, spring, summer, fall), drought timing index (drought_timing), flowering time (FT10), and minimum temperature (BIO6).
DOI: https://doi.org/10.7554/eLife.41038.011

• Supplementary file 2. Multiple linear regression model summaries. (A) Flowering time predicted by seasonal drought frequencies. *Arabidopsis* common garden flowering times were predicted by historic drought frequencies (DF) during different seasons at ecotypes' location of origin using multiple linear regression. (B) The strength of association between LoF alleles and flowering time ($-\log_{10}$ transformed P values) predicted by the strength of LoF alleles with drought timing, latitude, and minimum temperature.
DOI: https://doi.org/10.7554/eLife.41038.012

• Supplementary file 3. Genes. (A) Matrix of functional allele calls for 2088 genes among 1135 *Arabidopsis* ecotypes. LoF alleles are those with less than 90% predicted protein product and are classified with a '1'. Function alleles are classified with a '0'. (B) Associations between functional allele state and drought timing and flowering time for 2088 genes. Includes gene, estimate for logistic regression model testing the association between functional allele state and drought timing (Drought_timing_B) and flowering time (flw_10_B) after accounting for population structure, and the P-value of these estimates before Bonferroni correction for multiple testing (Drought_timing_p and flw_10_p). These values are also reported for LoF associations with latitude (lat_B, lat_p) and minimum temperature (temp_B, temp_p). (C) Selection statistics for 2088 genes. Includes $P_N/P_S$ (pnps), $D_N/D_S$ (dnds), frequency, number of LoF alleles, and average frequency per LoF allele. (D). Survey of sample genes with previously identified LoF alleles. (E) LoF alleles identified in previously studied genes (those surveyed in Table D). (F) Flowering time in T-DNA knockout lines. Flowering time (lsmean and standard error) of wild-type genomic background and T-DNA knockout lines of a sample of candidate genes in which LoF alleles are associated with spring drought environments or later flowering time phenotypes in *Arabidopsis* ecotypes.
DOI: https://doi.org/10.7554/eLife.41038.013

• Transparent reporting form

DOI: https://doi.org/10.7554/eLife.41038.014

## Data availability

All data generated or analyzed during this study are included in the manuscript and supporting files.

The following previously published datasets were used:

| Author(s) | Year | Dataset title | Dataset URL | Database and Identifier |
|---|---|---|---|---|
| The 1001 Genomes Consortium | 2016 | GMI-MPI Arabidopsis thaliana genomes | http://1001genomes.org/data/GMI-MPI/releases/ | 1001 Genomes Data Center, GMI-MPI |
| Kogan F | 1995 | Vegetative Health Index | https://www.star.nesdis.noaa.gov/smcd/emb/vci/VH/vh_ftp.php | National Oceanic and Atmospheric Administration, VHI |

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
