## [Decision Letter]

Thank you for submitting your article "Drought adaptation in *Arabidopsis thaliana* by extensive genetic loss-of-function" for consideration by *eLife*. Your article has been reviewed by three peer reviewers, one of whom is a member of our Board of Reviewing Editors, and the evaluation has been overseen by Christian Hardtke as the Senior Editor.

The reviewers have discussed the reviews with one another and the Reviewing Editor has drafted this decision to help you prepare a revised submission.

Summary:

This manuscript identifies a subset of existing natural knockouts and provides evidence that they are likely causal in natural variation within the species.

Essential revisions:

1) There was a concern that the inflated GWAS significance may be a result of some consistent error imparted by the functional allele assignment rather than all the genes being causal. Some analysis that can argue against this would be helpful especially to convince the generalist reviewer. We came up with two ideas but are willing to assess any other test you can develop. The ideas raised in our discussion were a) permute the phenotype to see that after this the glm will not lead to inflated results or b) permute the respective binary gene-wise score to see that an arbitrary assignment with the same allele frequency will agree with the null hypothesis.

2) The eco/eco aspects of the events needs to be assessed more broadly given the breadth of *Arabidopsis* life styles and how positive in one may be negative in another.

3) There is a need to better discuss the calling of specific events as a number of known events were not found.

Reviewer #1:

The authors conduct a survey to find LOF mutations with regards to the Col-0 reference genome. They then work to show that there is an association to potential adaptation and drought. This is a highly interesting manuscript but there are some issues with false negative rates in the LOF lists and referencing of the primary literature.

One conflict in this manuscript that I had was the idea that the whole manuscript was about drought adaptation yet the validation was on flowering time. There was no real discussion on if these mutants may or may not alter drought responses and if so, are those effects as unidirectional as for flowering. This conflict would optimally be resolved with experimental data at best or alternatively with discussion reflecting this difficulty.

I find it odd that none of the citations for LOF mutations contributing to adaptation or fitness are prior to 2006 even though there are a large number of *Arabidopsis* and other mutations that had LOF natural variation found prior to that. This includes key genes controlling flowering and defense such as RPM1, RPS2, FLM, AOP2, etc. Some of these genes such as the work by Bergelson on R genes and Kliebenstein on glucosinolates have direct evidence of field fitness effects of these natural variants in LOF. I understand that the authors prefer to use review articles but they should really use primary research literature as that is the real work that should be given acknowledgement especially as there is no length limit in *eLife*. This lack of primary literature may have led to the next issue about the LOF gene list.

A cursory analysis of the list of genes in the supplementary information found that the list is missing a number of genes with published loss of function events, I.e. BRX, AOP2, MAM, etc. This indicates that there is a significant false negative issue within the compilation of genes. The authors need to go through the literature to identify a collection of genes with known loss-of-function events and then assess how many they did or did not find. This is essential to let future researchers know how complete the list of genes is or is not. Is it possible that this is biased by use of the Col-0 genome as the reference and potentially not looking for GOF alleles in the other accessions which would be LOF if you shift the reference genome?

Equally, it seems like the authors should discuss settings where the LOF are not multiple independent events as is the case for RPM1 and RPS2. The general text has a feeling that all LOF are multiple independent events which may come from the soft sweep citation but that is not the exclusive view for plant natural variation.

Flowering time analysis seems to have only been conducted in one environment. The authors should discuss the fact that the environment has a key role in determining flowering time and how doing a broader range of environments with the mutants may influence the results.

For Figure 3F, is a linear correlation the best fit to the data? It looks like a non-linear correlation would be a better fit. The authors should do a model comparison of linear and non-linear regressions to see which best fits the data as a non-linear fit could alter the interpretation as that would suggest a maximal effect.

Reviewer #2:

This comprehensive study integrates across diverse approaches to detect drought timing and evaluate the genetic basis of adaptation to drought in the context of loss of function in the model organism, *Arabidopsis thaliana*. The innovative use of the Vegetative Health Index generated data on the timing of drought for numerous accessions of *Arabidopsis*. This approach could potentially be applied to other systems. The current study uses previously published genomic data to detect potential candidate genes associated with drought (as measured via the VHI) and flowering time (from a previously published growth chamber experiment). After evaluating statistical associations between drought, loss of function genes, and flowering time, the authors conducted gene knock out studies at several candidate genes showing relationships between loss of function and spring drought to evaluate causal link with flowering time.

I wonder about the adaptive nature of these associations. For example, is delayed flowering adaptive under spring drought and earlier flowering adaptive under summer drought? That is, are loss of function alleles associated with adaptive changes in flowering phenology? In the third paragraph of the Results and Discussion, the authors point to two studies (Kooyers, 2015; and Dittberner, 2018) to support the assertion that these phonological changes are adaptive. Unfortunately, the Dittberner Endnote citation was inadvertently excluded from the references, so that I cannot look at it. Kooyers, 2015, discusses drought avoidance vs. escape as general plant strategies, with escape associated with rapid growth and avoidance associated with other morphological and physiological traits that confer higher water-use efficiency. The typical thought is that plants can escape from drought by flowering early. In the current study, the authors suggest that later flowering genotypes may avoid spring drought. When does germination occur in sites with spring drought? Late flowering genotypes would still experience the spring drought as juveniles, depending on when germination occurred. It does not seem clear that delay flowering enables those plants to escape from the drought, given that early life history stages are very susceptible to drought. It seems problematic to refer to loss of function as generating adaptive shifts in flowering phenology without fitness data (ideally in the field) to test those hypotheses directly. That said, I appreciate that the Dn/Ds and Pn/Ps analyses point to positive selection for loss of function alleles in genes associated with drought or flowering time.

Figure 1A: What data are used to determine the regions of drought stress (in graded brown at the bottom of the top two panels)? The Materials and methods (subsection “Satellite-Detected Drought Histories of *Arabidopsis*”, first paragraph) set 40 as the threshold for drought (values of HVI <40 are indicative of drought). How was that value determined? How does it relate to drought stress as perceived or experienced by *Arabidopsis* in nature? The authors use the HVI to determine the timing of drought for *Arabidopsis*. Have they ground-truthed these drought metrics in any of the field sites? How reliable is the HVI for characterizing exposure of *Arabidopsis* to drought in its native range?

Figure 1C focuses on spring vs. summer droughts. Have winter droughts (present in panel 1B, right side) affected the timing of flowering of *Arabidopsis* in any of these populations? It seems like winter drought could affect flowering time for both fall germinating and spring germinating ecotypes.

What other factors could drive population divergence between populations with spring vs. summer drought? The manuscript seems to assume that drought is the only factor affecting those differences. For example, the Materials and methods state "Summer drought genes were identified as those in which LoF alleles are found in ecotypes that experience a significantly (β_drought timing_ <0 & P_drought timing_ <0.05) more negative drought-timing index […] Conversely, spring drought genes were identified as those in which LoF alleles are found in ecotypes that experience a significantly (_βdrought timing_ > 0 & P_drought timing_ <0.05) more positive drought-timing index…" Are there other environmental factors that covary with drought that could also influence evolution at these loci? How can the authors be sure that these are really "summer drought" vs. "spring drought" genes? Are these genes consistent with mapped regions for drought tolerance in *Arabidopsis*?

Additional points:

I recommend deleting the first part of the sentence ("Plants have been adapting to drought for millennia.…"). For one, plants have been adapting to drought ever since they colonized land from the mid-Ordovician to the Devonian, over 400 million years ago. Secondly, this phrase does not provide information that advances the narrative.

Introduction, second paragraph: Please provide citations for the statement that most research has focused on late-season droughts. This statement does not resonate with my experience conducting studies and reviewing manuscripts. When possible to manipulate in the field, researchers impose drought in an ecologically-relevant fashion. In the lab, researchers generally time drought treatments for a developmentally-relevant stage.

Figure 1A It might be useful to label the two locations with the names of the *Arabidopsis* accessions or provide the geographic region, in addition to the latitudes and longitude.

Both panels of Figure 1A (especially the panel on the right) seem to imply that drought stress is occurring less frequently through time. The darker lines indicative of more recent years seem to occur in regions of higher VHI. Is that correct?

In the subsection “Experimental Testing of Predicted Phenotypes in Gene Knockout Lines”, it states that flowering time was assessed as days from planting to the emergence of the first flower. Is there variation in germination timing? Why not measure flowering time as days from germination to the first flower?

Figure 3F: Is this relationship linear or might a curvilinear model fit better?

Reviewer #3:

Monroe and colleagues describe the link of loss-of-function Alleles with drought adaptation and flowering time in *A. thaliana*. The manuscript is well written and interesting conclusions are reported. Especially the high overlap of associations for summer drought and early flowering and spring drought and late flowering is intriguing. Additional the functional follow-up in T-DNA knock out lines is excellent.

Still, I have one major comment.

My major concern is the statistical framework used for GWAS.

The authors used logistic regression in a glm and added the first 3 principle components to correct for population structure. This differs from the standard GWAS procedure in *A. thaliana* which uses a linear mixed model to correct for population structure confounding. The rational why the authors used this model is not well described in the manuscript. Additionally, the results differ markedly from the analysis with a classical LMM. (I run a normal LMM with the provided data for comparison, happy to provide this if needed) Next, the qq_plot is also highly inflated (which might be expected collapsing LoF Alleles to one score per gene), but is not if a normal LMM is used.

To summarize, I am not completely sure what to make out of this, especially as the results and conclusion look really nice with the presented method (e.g. Figure 2 is really impressive).

Still, it would be good if the authors at least comment on why to use the proposed framework and the inflation observed in a qq_plot.

---

## [Author Response]

Essential revisions:1) There was a concern that the inflated GWAS significance may be a result of some consistent error imparted by the functional allele assignment rather than all the genes being causal. Some analysis that can argue against this would be helpful especially to convince the generalist reviewer. We came up with two ideas but are willing to assess any other test you can develop. The ideas raised in our discussion were a) permute the phenotype to see that after this the glm will not lead to inflated results or b) permute the respective binary gene-wise score to see that an arbitrary assignment with the same allele frequency will agree with the null hypothesis.

Thank you for expressing these concerns. We too wanted to verify that the results were not an artifact of the functional allele assignment and greatly appreciate the ideas to address this.

The revised manuscript includes new analyses inspired by this suggestion. We chose to implement a permutation method based on idea (b) above, permuting the genotype matrix but keeping allele frequencies identical. We then repeated the genome wide association scan and compared the results to those generated by the natural genotype matrix. By permuting the genotype matrix we were able to address 1) the concerns about the allele assignments explaining the inflated significance as well as 2) the possibility that the overlap of gene associations with drought timing and flowering time being simply explained by the correlation between ecotype flowering times and drought timing at their home environments.

As expected, we found that for the permuted genotype matrix, p values were not inflated and fell within the confidence interval of the expected observed=expected line. Indeed, when we permuted the genotype matrix, we found no genes that were significantly associated with drought timing or flowering time after a Bonferroni correction. This result has been added to a new supplementary figure in the manuscript (Figure 2—figure supplement 1).

Additionally, we found that the correlation between p values for gene associations with drought timing and p values for gene associations to flowering time fell from 0.48 to 0.01 in the permuted genotypes. Because drought timing and flowering time vectors were unmodified, this indicates that the overlaps we observed between gene associations are not explained entirely by collinearity between drought timing and flowering time. This result has also been included in Figure 2—figure supplement 1.

Finally, we added a section to the Materials and methods with a more detailed explanation of the rationale behind our functional genome wide association scanning approach. Specifically, we discuss why the tests are more likely to point to phenotypically impactful genetic variation and why a functional genome wide association approach places greater priority on reducing false negative associations than traditional GWAS. We hope that together, these new results and the more thorough explanation of the methods, provide readers with a better understanding of the primary findings. Further details are provided in the responses to specific reviewer concerns below.

2) The eco/eco aspects of the events needs to be assessed more broadly given the breadth of Arabidopsis life styles and how positive in one may be negative in another.

Thank you for the suggestion. We agree that the initial manuscript failed to adequately consider the results in the context of *Arabidopsis* ecology and life history. The revised manuscript contains numerous changes to better frame this work around the natural ecology of *Arabidopsis*. For example:

“Flowering time is only one component of phenology and other adaptive life history transitions such a germination timing (Donohue 2002) may also be influenced by drought timing and could change how drought timing affects the evolution of flowering time, a hypothesis that warrants further investigation.”

To the revised manuscript, we have also added several new analyses that we hope will yield address some reviewer concerns about the conclusions drawn from the findings presented in the original submission. Specifically, we present two new multiple linear regressions showing that spring and summer drought frequency are the most important predictors of flowering time, rather than drought frequency during other seasons, and that drought timing is a good predictor of allele associations to flowering time even while accounting for allele associations to other potentially important environmental variables, latitude and minimum temperature. These results are included in two supplemental tables (Supplementary file 3A, B). Further details are provided in the responses to specific reviewer concerns below.

3) There is a need to better discuss the calling of specific events as a number of known events were not found.

Thank you for bringing up this important point. The revised manuscript now contains a survey of LoF alleles in previously studied genes using our calling method and with the findings reported in new table (Supplementary file 2D, E). We confirmed the presence of LoF alleles in all of these genes except aop2, which has no gene model in Col-0 because it is annotated as a pseudogene and could not be evaluated with our pipeline. Most of these previously known LoF containing genes however, were not included in the 2088 which we tested for associations to drought timing and flowering time because the LoF allele frequencies were below our filtering threshold. The revised manuscript now contains the following paragraph:

“It should be noted that the 2088 genes tested for associations to flowering time and drought timing are not a complete representation of LoF alleles in *Arabidopsis*. […] Thus, while the methods used here are designed to minimize false positives (alleles classified as LoF, but which are actually functional), the likely occurrence of false negatives (undetected LoF alleles) in available data motivates the need for more sophisticated species wide genome sequencing efforts including a greater diversity of de-novo quality genomes for comprehensive detection of functionally relevant genetic variation across the species.”

Further details are provided in the responses to specific reviewer concerns below.

Reviewer #1:The authors conduct a survey to find LOF mutations with regards to the Col-0 reference genome. They then work to show that there is an association to potential adaptation and drought. This is a highly interesting manuscript but there are some issues with false negative rates in the LOF lists and referencing of the primary literature.One conflict in this manuscript that I had was the idea that the whole manuscript was about drought adaptation yet the validation was on flowering time. There was no real discussion on if these mutants may or may not alter drought responses and if so, are those effects as unidirectional as for flowering. This conflict would optimally be resolved with experimental data at best or alternatively with discussion reflecting this difficulty.

Thank you for raising this point. We have added the following statement to express this need for further experimental data:

“Future experimental work will be valuable to identify other plant physiological traits affected by the LoF alleles associated with drought timing.”

We have also revised the manuscript to clarify the connection between drought timing and flowering time in several places.

In revised manuscript:

“Flowering time in *Arabidopsis* is correlated with other drought tolerance traits such as water use efficiency and can serve as a proxy for alternative drought tolerance strategies, with early flowering genotypes being associated with low water use efficiency (drought escape strategy) and late flowering genotypes with high water use efficiency (dehydration avoidance strategy) (McKay et al., 2003; Lovell et al., 2013; Kenney et al., 2014). […] This hypothesis motivated our investigation to identify alleles associated with drought timing and test the prediction that they contribute to adaptive flowering time evolution.”

In revised manuscript:

“These results further support the classical hypothesis that the relationship between phenology and drought timing is the most important feature of plant drought tolerance (Passioura, 1996), indicating the evolution of “drought escape” through earlier flowering in summer drought environments, and “dehydration avoidance” by later flowering genotypes in spring drought environments. […]This pattern is also consistent with hypotheses explaining the more water conservative water use and stomatal traits observed in late flowering genotypes (Kooyers, 2015)(McKay et al., 2003; Lovell et al., 2013; Kenney et al., 2014) and those from spring drought environments (Dittberner et al., 2018).”

I find it odd that none of the citations for LOF mutations contributing to adaptation or fitness are prior to 2006 even though there are a large number of Arabidopsis and other mutations that had LOF natural variation found prior to that. This includes key genes controlling flowering and defense such as RPM1, RPS2, FLM, AOP2, etc. Some of these genes such as the work by Bergelson on R genes and Kliebenstein on glucosinolates have direct evidence of field fitness effects of these natural variants in LOF. I understand that the authors prefer to use review articles but they should really use primary research literature as that is the real work that should be given acknowledgement especially as there is no length limit in eLife. This lack of primary literature may have led to the next issue about the LOF gene list.

Thank you, this is an excellent point. These papers have been an inspiration and provide important background for this work. The revised manuscript now includes 32 citations for primary literature studies of adaptive LoF in *Arabidopsis*.

In revised manuscript:

“Indeed, a number of individual genes exhibiting evidence of locally adaptive loss-of-function have been documented in *Arabidopsis* (Grant et al., 1998; Johanson et al., 2000; Kliebenstein et al., 2001; Kroymann et al., 2003; Mouchel et al., 2004)(Aukerman et al., 1997; Hauser et al., 2001; Mauricio et al., 2003; Alonso-Blanco et al., 2005; Werner et al., 2005; Barboza et al., 2013; Xiang et al., 2014).”

Additional citations are found in the new table (Supplementary file 3D) which contains a survey of previously identified LoF mutants in *Arabidopsis*.

A cursory analysis of the list of genes in the supplementary information found that the list is missing a number of genes with published loss of function events, I.e. BRX, AOP2, MAM, etc. This indicates that there is a significant false negative issue within the compilation of genes. The authors need to go through the literature to identify a collection of genes with known loss-of-function events and then assess how many they did or did not find.

Thank you, this is a great suggestion. The revised manuscript includes a survey of previously studied LoF mutants (Supplementary file 3D).

The revised manuscript also discusses the issue of false negatives more directly.

In revised manuscript:

“It should be noted that the 2088 genes tested for associations to flowering time and drought timing are not a complete representation of LoF alleles in *Arabidopsis*. […] Thus, while the methods used here are designed to minimize false positives (alleles classified as LoF, but which are actually functional), the likely occurrence of false negatives (undetected LoF alleles) in available data motivates the need for more sophisticated species wide genome sequencing efforts including a greater diversity of de-novo quality genomes for comprehensive detection of functionally relevant genetic variation across the species.”

This is essential to let future researchers know how complete the list of genes is or is not. Is it possible that this is biased by use of the Col-0 genome as the reference and potentially not looking for GOF alleles in the other accessions which would be LOF if you shift the reference genome?

We agree. See quote above for points about the potential sources of false negatives (including LOF in Col-0) and the need to develop multiple high quality reference genomes to better study functional genetic variation at genomic scales.

In revised manuscript:

“Because the reference genome and gene models are from an early flowering *Arabidopsis* line, Col-0, this is consistent with the hypothesis that LoF alleles are particularly important in the evolution of phenotypic divergence (Rausher 2008). This result also highlights the need to develop functional genomics resources informed by multiple de-novo quality reference genomes.”

In revised manuscript, Abstract

“These results also motivate improved species-wide sequencing efforts to better identify loss-of-function variants”

Equally, it seems like the authors should discuss settings where the LOF are not multiple independent events as is the case for RPM1 and RPS2. The general text has a feeling that all LOF are multiple independent events which may come from the soft sweep citation but that is not the exclusive view for plant natural variation.

We agree that it is important to also consider cases with single LoF allele. See excerpt from the revised manuscript reflecting on this idea below.

In revised manuscript:

“In cases where adaptation proceeds through the fixation of a single adaptive allele, traditional genome scanning approaches may be sufficient to detect causal loci. However, when genetic variation consists of multiple independent alleles, as is often the case for the genes examined here (Figure 2—figure supplement 2), classifying alleles functionally before testing for associations is likely necessary.”

Flowering time analysis seems to have only been conducted in one environment. The authors should discuss the fact that the environment has a key role in determining flowering time and how doing a broader range of environments with the mutants may influence the results.

Agreed. See text below:

In revised manuscript:

“Furthermore, measuring flowering time in other environments, such alternate light regimes, may yield a different set of candidate genes using similar approaches.”

For Figure 3F, is a linear correlation the best fit to the data? It looks like a non-linear correlation would be a better fit. The authors should do a model comparison of linear and non-linear regressions to see which best fits the data as a non-linear fit could alter the interpretation as that would suggest a maximal effect.

We agree that the scatterplot appears to have a non-linear trend. However, adding a non-linear predictor did not improve the model fit. We have added this result to the revised manuscript.

In revised manuscript:

“We found that flowering time is strongly predicted by the accumulation of LoF alleles across the 214 candidate genes associated to spring drought and/or later flowering time (Figure 3A-E), estimating a 1-day increase for every 3 additional LoF alleles across these candidate genes (Figure 3F). This relationship is best represented as a simple linear regression; the addition of a non-linear quadratic predictor variable did not significantly improve the fit of the model (F= 0.7005, P = 0.4028).”

Reviewer #2:This comprehensive study integrates across diverse approaches to detect drought timing and evaluate the genetic basis of adaptation to drought in the context of loss of function in the model organism, Arabidopsis thaliana. The innovative use of the Vegetative Health Index generated data on the timing of drought for numerous accessions of Arabidopsis. This approach could potentially be applied to other systems. The current study uses previously published genomic data to detect potential candidate genes associated with drought (as measured via the VHI) and flowering time (from a previously published growth chamber experiment). After evaluating statistical associations between drought, loss of function genes, and flowering time, the authors conducted gene knock out studies at several candidate genes showing relationships between loss of function and spring drought to evaluate causal link with flowering time.I wonder about the adaptive nature of these associations. For example, is delayed flowering adaptive under spring drought and earlier flowering adaptive under summer drought? That is, are loss of function alleles associated with adaptive changes in flowering phenology? In the third paragraph of the Results and Discussion, the authors point to two studies (Kooyers, 2015; and Dittberner, 2018) to support the assertion that these phonological changes are adaptive. Unfortunately, the Dittberner Endnote citation was inadvertently excluded from the references, so that I cannot look at it. Kooyers, 2015, discusses drought avoidance vs. escape as general plant strategies, with escape associated with rapid growth and avoidance associated with other morphological and physiological traits that confer higher water-use efficiency. The typical thought is that plants can escape from drought by flowering early. In the current study, the authors suggest that later flowering genotypes may avoid spring drought. When does germination occur in sites with spring drought? Late flowering genotypes would still experience the spring drought as juveniles, depending on when germination occurred. It does not seem clear that delay flowering enables those plants to escape from the drought, given that early life history stages are very susceptible to drought. It seems problematic to refer to loss of function as generating adaptive shifts in flowering phenology without fitness data (ideally in the field) to test those hypotheses directly.

Thank you for raising these concerns. The revised manuscript includes several new sections inspired by the points brought up here. We agree that it is extremely challenging to demonstrate the adaptive value of genetic variation, but hope that the revised manuscript provides a clearer picture of the hypotheses about drought adaptation that we are aiming to address. Here are some examples:

In revised manuscript:

“Flowering time in *Arabidopsis* is correlated with other drought tolerance traits such as water use efficiency and can serve as a proxy for alternative drought tolerance strategies, with early flowering genotypes being associated with low water use efficiency (drought escape strategy) and late flowering genotypes with high water use efficiency (dehydration avoidance strategy) (McKay et al., 2003; Lovell et al., 2013; Kenney et al., 2014). […] This hypothesis motivated our investigation to identify alleles associated with drought timing and test the prediction that they contribute to adaptive flowering time evolution.”

In revised manuscript:

“These results further support the classical hypothesis that the relationship between phenology and drought timing is the most important feature of plant drought tolerance (Passioura, 1996), indicating the evolution of “drought escape” through earlier flowering in summer drought environments, and “dehydration avoidance” by later flowering genotypes in spring drought environments. Because most *Arabidopsis* populations appear to exhibit a winter annual life habit, germinating in the fall and overwintering as a rosette (Ratcliffe, 1961; Thompson 1994; Burghardtet al., 2015), late flowering genotypes in spring drought environments are expected to still encounter drought conditions. […] This pattern is also consistent with hypotheses explaining the more water conservative water use and stomatal traits observed in late flowering genotypes (Kooyers, 2015)(McKay et al., 2003; Lovell et al., 2013; Kenney et al., 2014) and those from spring drought environments (Dittberner et al., 2018).”

That said, I appreciate that the Dn/Ds and Pn/Ps analyses point to positive selection for loss of function alleles in genes associated with drought or flowering time.Figure 1A: What data are used to determine the regions of drought stress (in graded brown at the bottom of the top two panels)? The Materials and methods (subsection “Satellite-Detected Drought Histories of Arabidopsis”, first paragraph) set 40 as the threshold for drought (values of HVI <40 are indicative of drought). How was that value determined? How does it relate to drought stress as perceived or experienced by Arabidopsis in nature? The authors use the HVI to determine the timing of drought for Arabidopsis. Have they ground-truthed these drought metrics in any of the field sites? How reliable is the HVI for characterizing exposure of Arabidopsis to drought in its native range?

Great questions. The definition of drought as VHI below 40 was determined by models used by the developers of the VHI at NOAA. We have made several changes in the revised manuscript that we hope will provide some clarification.

In revised manuscript:

“(B) drought frequency (VHI<40, NOAA drought classification) by week (line) and season (bars).”

In revised manuscript:

“One such measurement, the Vegetative Health Index (VHI) has been used for decades to monitor drought, including in many places across the natural range of *Arabidopsis* (Kogan, 1997).”

Figure 1C focuses on spring vs. summer droughts. Have winter droughts (present in panel 1B, right side) affected the timing of flowering of Arabidopsis in any of these populations? It seems like winter drought could affect flowering time for both fall germinating and spring germinating ecotypes.

This is a good point. The revised manuscript has an additional analysis (multiple linear regression) to address this. We found that only spring and summer drought frequencies are significant predictors of flowering time. (Supplementary file 2A).

What other factors could drive population divergence between populations with spring vs. summer drought? The manuscript seems to assume that drought is the only factor affecting those differences. For example, the Materials and methods state "Summer drought genes were identified as those in which LoF alleles are found in ecotypes that experience a significantly (β_drought timing_ <0 & P_drought timing_ <0.05) more negative drought-timing index [..] Conversely, spring drought genes were identified as those in which LoF alleles are found in ecotypes that experience a significantly (_βdrought timing_ > 0 & P_drought timing_ <0.05) more positive drought-timing index" Are there other environmental factors that covary with drought that could also influence evolution at these loci? How can the authors be sure that these are really "summer drought" vs. "spring drought" genes? Are these genes consistent with mapped regions for drought tolerance in Arabidopsis?

Another good point. We created an additional multiple linear regression approach to address this. Specifically, we tested whether flowering time allele associations were predicted by drought timing allele associations while controlling for the associations between alleles and both latitude and minimum temperature, two variables that could also drive flowering time evolution. Nevertheless, we recognize that other factors could (likely) explain some of the variance in the distribution flowering time alleles. See some relevant excerpts from the revised manuscript below to address this point and Supplementary file 2B:

In revised manuscript:

“These results provide new insight into the ecology and genetics of *Arabidopsis* life history evolution, but the complex ecological reality of these processes is undoubtedly beyond the scope of this study. […] However, other unknown climatic variables or environmental interactions and non-linearities likely contribute to the flowering time adaptation as well.”

Additional points:I recommend deleting the first part of the sentence ("Plants have been adapting to drought for millennia.…"). For one, plants have been adapting to drought ever since they colonized land from the mid-Ordovician to the Devonian, over 400 million years ago. Secondly, this phrase does not provide information that advances the narrative.

Thank you for the feedback. We have removed this line and the paragraph now begins with:

In revised manuscript:

“Drought stress can occur throughout the year and drought timing is forecast to change over the next century (Trenberth et al., 2014). While dramatic evolutionary responses to drought events have been documented, (e.g. Franks et al., 2007), little is known about the relationship between drought timing and adaptation.”

Introduction, second paragraph: Please provide citations for the statement that most research has focused on late-season droughts. This statement does not resonate with my experience conducting studies and reviewing manuscripts. When possible to manipulate in the field, researchers impose drought in an ecologically-relevant fashion. In the lab, researchers generally time drought treatments for a developmentally-relevant stage.

We have removed this statement.

Figure 1A It might be useful to label the two locations with the names of the Arabidopsis accessions or provide the geographic region, in addition to the latitudes and longitude.

Fixed.

Both panels of Figure 1A (especially the panel on the right) seem to imply that drought stress is occurring less frequently through time. The darker lines indicative of more recent years seem to occur in regions of higher VHI. Is that correct?

We analyzed the data and found no significant increase in the frequency of drought at these two locations. However, this is interesting observation that might warrant a thorough investigation. Indeed, this data presents lots of opportunities for future work.

In the subsection “Experimental Testing of Predicted Phenotypes in Gene Knockout Lines”, it states that flowering time was assessed as days from planting to the emergence of the first flower. Is there variation in germination timing? Why not measure flowering time as days from germination to the first flower?

This is a good point. As we now report in the Materials and methods section, we chose to measure flowering time based on the definition used by the 1,001 Genomes Consortium.

Figure 3F: Is this relationship linear or might a curvilinear model fit better?

We agree that the scatterplot appears to have a non-linear trend. However, adding a non-linear predictor did not improve the model fit. We have added this result to the revised manuscript.

In revised manuscript:

“We found that flowering time is strongly predicted by the accumulation of LoF alleles across the 214 candidate genes associated to spring drought and/or later flowering time (Figure 3A-E), estimating a 1-day increase for every 3 additional LoF alleles across these candidate genes (Figure 3F). This relationship is best represented as a simple linear regression; the addition of a non-linear quadratic predictor variable did not significantly improve the fit of the model (F= 0.7005, P = 0.4028).”

Reviewer #3:Monroe and colleagues describe the link of loss-of-function Alleles with drought adaptation and flowering time in A. thaliana. The manuscript is well written and interesting conclusions are reported. Especially the high overlap of associations for summer drought and early flowering and spring drought and late flowering is intriguing. Additional the functional follow-up in T-DNA knock out lines is excellent.Still, I have one major comment.My major concern is the statistical framework used for GWAS.The authors used logistic regression in a glm and added the first 3 principle components to correct for population structure. This differs from the standard GWAS procedure in A. thaliana which uses a linear mixed model to correct for population structure confounding. The rational why the authors used this model is not well described in the manuscript. Additionally, the results differ markedly from the analysis with a classical LMM. (I run a normal LMM with the provided data for comparison, happy to provide this if needed) Next, the qq_plot is also highly inflated (which might be expected collapsing LoF Alleles to one score per gene), but is not if a normal LMM is used.To summarize, I am not completely sure what to make out of this, especially as the results and conclusion look really nice with the presented method (e.g. Figure 2 is really impressive).Still, it would be good if the authors at least comment on why to use the proposed framework and the inflation observed in a qq_plot.

Thank you for sharing your thoughts. To the revised manuscript, we have added a section to the Materials and methods that we hope will clarify the rationale behind our method. We have also added additional analyses relevant for interpreting the qqplot results. Specifically, we permuted the genotype matrix and repeated the association analyses, to see if allele assignments explained the inflated results. Here are a few excerpts from the revised manuscript to inspired by the concerns you raised.

In revised manuscript:

“This association study differs from traditional GWAS in several respects. […] For these reasons, we implemented analyses based on (Price et al., 2006) to balance false positives and false negatives.”

In revised manuscript:

“The preceding analyses revealed considerable overlap between genes associated with both drought timing and flowering time. […] Quantile-quantile plots of P values were visualized using qqPlot in the GWASTools package in R (Gogarten et al., 2012) (Figure 2—figure supplement 1A-D)”

In revised manuscript:

“To control for the possibility that allele frequencies or the relationship between drought timing and flowering time explained these observations, we also tested whether allele associations were correlated when generated from association analyses using a matrix of randomly permuted genotypes with the same allele frequencies (Figure 2—figure supplement 1F, r^2^ = 0.01).”

In revised manuscript:

“After filtering to reduce the likelihood of false positives (see Materials and methods), we thus tested 2088 genes for LoF allele associations with drought timing (Figure 2A) and flowering time (Figure 2B). […] In contrast, when we performed these analyses on a permuted LoF genotype matrix, we found no genes that were significantly associated with drought timing or flowering time (Figure 2—figure supplement 1B, D).”

In revised manuscript:

“The strengths of the associations between LoF alleles and drought timing (P values) was also strongly correlated with the strengths of the associations to flowering time (r^2^ = 0.48 Figure 2—figure supplement 1E, Figure 2C, D). […] In contrast, these associations were weakly correlated when genotypes were permuted (r^2^ = 0.01, Figure 2—figure supplement 1F), indicating that the result is not simply explained as an artifact of allele frequencies or by the relationship between drought timing and flowering time.”